# Seroprevalence of antibodies against *Chlamydia trachomatis* and enteropathogens and distance to the nearest water source among young children in the Amhara Region of Ethiopia

Kristen Aiemjoy[1]*, Solomon Aragie[2], Dionna M. Wittberg[3], Zerihun Tadesse[2], E. Kelly Callahan[4], Sarah Gwyn[5], Diana Martin[5], Jeremy D. Keenan[3], Benjamin F. Arnold[3]

**1** Division of Infectious Diseases and Geographic Medicine, Stanford University School of Medicine, Stanford, California, United States of America, **2** The Carter Center, Addis Ababa, Ethiopia, **3** Francis I. Proctor Foundation, University of California San Francisco, San Francisco, California, United States of America, **4** The Carter Center, Atlanta, Georgia, United States of America, **5** U.S. Centers for Disease Control and Prevention, Atlanta, Georgia, United States of America

* kaiemjoy@stanford.edu

**Data Availability Statement:** The data and codebook are available on the Open Science

## Abstract

The transmission of trachoma, caused by repeat infections with *Chlamydia trachomatis*, and many enteropathogens are linked to water quantity. We hypothesized that children living further from a water source would have higher exposure to *C. trachomatis* and enteric pathogens as determined by antibody responses. We used a multiplex bead assay to measure IgG antibody responses to *C. trachomatis*, *Giardia intestinalis*, *Cryptosporidium parvum*, *Entamoeba histolytica*, *Salmonella enterica*, *Campylobacter jejuni*, enterotoxigenic *Escherichia coli* (ETEC) and *Vibrio cholerae* in eluted dried blood spots collected from 2267 children ages 0–9 years in 40 communities in rural Ethiopia in 2016. Linear distance from the child's house to the nearest water source was calculated. We derived seroprevalence cutoffs using external negative control populations, if available, or by fitting finite mixture models. We used targeted maximum likelihood estimation to estimate differences in seroprevalence according to distance to the nearest water source. Seroprevalence among 1–9-year-olds was 43% for *C. trachomatis*, 28% for *S. enterica*, 70% for *E. histolytica*, 54% for *G. intestinalis*, 96% for *C. jejuni*, 76% for ETEC and 94% for *C. parvum*. Seroprevalence increased with age for all pathogens. Median distance to the nearest water source was 473 meters (IQR 268, 719). Children living furthest from a water source had a 12% (95% CI: 2.6, 21.6) higher seroprevalence of *S. enterica* and a 12.7% (95% CI: 2.9, 22.6) higher seroprevalence of *G. intestinalis* compared to children living nearest. Seroprevalence for *C. trachomatis* and enteropathogens was high, with marked increases for most enteropathogens in the first two years of life. Children living further from a water source had higher seroprevalence of *S. enterica and G. intestinalis* indicating that improving access to water in the Ethiopia's Amhara region may reduce exposure to these enteropathogens in young children.

Framework (https://osf.io/srdqw/) (DOI 10.17605/OSF.IO/SRDQW).

**Funding:** This study was funded by the National Institutes of Health, National Eye Institute (NEI U10 EY016214) (PI: Jeremy Keenan). The funders had no role in study design, data collection and analysis, decision to publish, or preparation of the manuscript.

**Competing interests:** The authors have declared that no competing interests exist.

## Author summary

Trachoma, an infection of the eye caused by the bacteria *Chlamydia trachomatis*, and many diarrhea-causing infections are associated with access to water for washing hands and faces. Measuring these different pathogens in a population is challenging and rarely are multiple infections measured at the same time. Here, we used an integrated approach to simultaneously measure antibody responses to *C. trachomatis*, *Giardia intestinalis*, *Cryptosporidium parvum*, *Entamoeba histolytica*, *Salmonella enterica*, *Campylobacter jejuni*, enterotoxigenic *Escherichia coli* (ETEC) and *Vibrio cholerae* among young children residing in rural Ethiopia. We found that the seroprevalence of all pathogens increased with age and that seropositivity to more than one pathogen was common. Children living further from a water source were more likely to be exposed to *S. enterica* and *G. intestinalis*. Integrated sero-surveillance is a promising avenue to explore the complexities of multi-pathogen exposure as well as to investigate associations between water, sanitation, and hygiene related exposures and disease transmission.

## Introduction

Diarrhea and trachoma typically afflict the world's poorest populations and are major contributors to preventable morbidity [1,2]. Diarrhea, caused by parasitic, viral and bacterial infections, and trachoma, caused by repeated *Chlamydia trachomatis* infections of the eye, share water and hygiene related transmission pathways. Increased access to water for food preparation and washing of hands, faces, and clothing is hypothesized to reduce transmission of both infectious diarrhea and *C. trachomatis* [3–6]. In regions where water must be carried from the source to the household, distance to the nearest water source will likely influence the quantity of water a household uses [7–10].

Antibody responses may be an informative and efficient approach to simultaneously measure enteropathogen and *C. trachomatis* exposure [11–13]. Unlike pathogen detection from stool samples or conjunctival swabs, antibody response integrates information over time, offering a longer window to identify exposed individuals. [12]. This advantage is especially desirable for studies with infrequent monitoring and data collection visits. Antibody response enumerates symptomatic, asymptomatic and past infections, revealing a more complete picture of transmission [12]. With recent advances in microsphere-based multiplex immunoassays, antibodies against multiple antigens can be detected simultaneously from a single blood spot [14]. This technology has a unique advantage that it can be used to simultaneously monitor for dozens of markers of pathogen transmission, potentially revealing vulnerable populations and/or individuals who experience the pervasive burdens of multiple-pathogen exposure.

In this study we evaluated IgG antibody responses to a panel of antigens from viral, bacterial, and protozoan enteropathogens and *C. trachomatis* antigens among a population-based cohort of children aged 0 to 9 years in rural Ethiopia. Our objectives were to describe age-dependent seroprevalence and co-prevalence of the pathogens and to evaluate if seroprevalence varied according to distance to nearest water source.

## Methods

### Ethics statement

Ethical approval for this study was granted by the National Research Ethics Review Committee of the Ethiopian Ministry of Science and Technology, the Ethiopian Food, Medicine, and

Health Care Administration and Control Authority, and institutional review boards at the University of California, San Francisco and Emory University. CDC staff did not have contact with study participants or access to personal identifying information and were therefore determined to be non-engaged. Community leaders provided verbal consent before enrollment of the community in the trial. Oral consent was approved by all the institutional review boards and was obtained from each participant or their guardian for participants younger than 18 years.

## Study design overview

We conducted a cross-sectional study evaluating antibody responses in children at the baseline visit of a cluster-randomized trial of a water, sanitation and hygiene (WASH) intervention in 40 communities (the cluster unit) in the Amhara region of Ethiopia. We used a multiplex bead assay to simultaneously measure IgG antibodies to antigens from *Chlamydia trachomatis* (Pgp3, CT694), *Giardia intestinalis (*VSP3, VSP5*)*, *Cryptosporidium parvum* (Cp17, Cp23), *Entamoeba histolytica* (LecA), *Salmonella enterica* (LPS Groups B and D), *Campylobacter jejuni* (p18, p39), enterotoxigenic *Escherichia coli* (ETEC heat labile toxin β subunit) and *Vibrio cholerae* (CtxB) from blood spots collected during the baseline study visit.

## Study population

Sanitation, Water, and Instruction in Face-washing for Trachoma (SWIFT), is an ongoing NIH-funded cluster-randomized trial designed to determine the effectiveness of a comprehensive WASH package for ocular *C. trachomatis* infection (NEI U10 EY016214) in three *woredas* (districts) of the Wag Hemra zone of Amhara, Ethiopia. Most of the rainfall in the Wag Hemra zone occurs in June, July and August, however there is significant seasonal and interannual variability, predisposing the region to drought [15]. The topography is mountainous with steep gorges and valleys. This cross-sectional analysis was carried out during the baseline study visit for SWIFT, from January to April 2016.

Study staff performed a door-to-door census in December 2015, approximately one month before the baseline examination visit began. Census workers recorded the name, sex, and age of each household member and the GPS coordinates of the house (accuracy of GPS approximately 15-20m). Age was calculated from the date of birth if known or the child's age in years for children older than one year and in months for children one year old and under." From this census, we drew a random sample of 30 children aged 0 to 5 years and 30 children aged 6 to 9 years in each cluster for inclusion in the study. The sample size was calculated for the primary outcome of the trial (molecular detection of ocular *C. trachomatis* infection).

## Measurements

**Dried blood spots.** A few days before each study visit a volunteer was sent out into the community to mobilize sampled children and their accompanying caregivers to attend the examination visit, with information on the time and location of the event. A trained laboratory technician lanced the index finger of each child and collected 5 blood spots onto a TropBio filter paper (Cellabs Pty Ltd., Brookvale, New South Wales, Australia) calibrated to hold 10 μL of blood per spot. The filter paper was labeled with a random number identification sticker, air-dried for at least one hour and then individually packaged in plastic re-sealable bags. The individual bags from each cluster were placed in large, re-sealable bags with desiccant. The samples were stored at -20˚C until all sample collection for the entire study visit was completed and then shipped at ambient temperature to the Centers for Disease Control and Prevention

(CDC) in Atlanta, GA, where they were stored at -20˚C until testing between February and March of 2017, approximately 12 months after collection.

**Distance to water.** At the time of the census, census workers asked community leaders to list all sources of water used in the community. The census workers then visited each water source, recorded the GPS coordinates and described the type of water source. Census workers were accompanied by the community leader or a community representative. Linear distance to the nearest water source was calculated from the household using GPS coordinates. We hypothesized that the quantity of water available to the household would have a larger effect on *C. trachomatis* and enteropathogen transmission compared to water quality, and thus used distance to the nearest water source (improved or unimproved) for the analysis. We calculated community-level distance to water as the median distance from each household in the community to its nearest water source.

**Covariates.** In a random one-third of households, study field workers performed a household survey evaluating socioeconomic status, access to water, number/type of animals in household, hygiene behaviors, and sanitation infrastructure. The survey was limited to a subset of households for budgetary reasons; additional details on the household survey are available elsewhere [16]. Distance to the nearest water source was calculated in the same way as above for the subset of households with the household survey.

## Laboratory methods

We measured IgG responses against *C. trachomatis* and enteropathogen antigens using a multiplex SeroMAP microsphere-based immunoassay on the Luminex xMAP platform (Luminex Corp, Austin, TX) for the following antigens: *G. intestinalis* variant-specific surface protein AS8/GST fusion (VSP3) and 42e/GST fusion (VSP5) [17–19]; *C. jejuni antigen* p39 and p18 [20–23]; *Enterotoxigenic Escherichia coli* (ETEC heat labile toxin B subunit)[12,24,25]; *C. parvum* 17-kDa protein/GST fusion (Cp17) and 23-kDa protein/GST fusion (Cp23)[26–30]; *Salmonella spp. (*LPS Groups B and D) [12,13,31,32]; *V. cholerae* toxin B subunit (CtxB); *E. histolytica* Gal/GalNAc lectin heavy chain subunit (LecA)[33–35]; and *C. trachomatis* Pgp3 & CT694 [36,37]. LPS B and D (Sigma Chemical, St. Louis, MO) were dissolved in 50 mM 2-(N-morpholino)ethanesulfonic acid (MES) at pH 5 with 0.1% 3-[(3-cholamidopropyl)dimethy-lammonio]-1-propanesulfonate (CHAPS) at a concentration of 1 mg/ml. Coupling reactions were conducted in 50 MES and 0.85% NaCl at pH 5 using 10 micrograms LPS/ 1.25 x 107 beads. The enteropathogens were selected on the basis of antigen availability and known circulation in the region. <u>Serum elution</u>: The dried blood spots were brought to room temperature and submerged in 1600 μL of elution buffer for a minimum of 18 hours at 4˚C [38]. <u>Multiplex bead assay</u>: *Each* 96-well plate included a buffer-only blank, one negative control, and two positive controls. The two positive control wells contained pooled serum that was previously classified as seropositive for each antigen at two dilutions: 1:100 and 1:1000. The background from the buffer-only blank is subtracted from the result for each antigen, and values are reported as an average median fluorescence intensity with background subtracted (MFI-bg) [39,40].

## Statistical analysis

For pathogens with two antigens (*C. trachomatis*, *G. intestinalis*, *C. parvum*, *C. jejuni* and *S. enterica*), children positive to either antigen were considered exposed.

Positivity cutoffs were defined using external control populations when available. For *C. trachomatis* Pgp3 & CT694 cutoffs were derived using ROC curves [38], for *C. parvum* Cp17 & Cp23 cutoffs were derived using a standard curve and for *G. intestinalis* VSP-3 & VSP-5 and *E. histolytica* LecA cutoffs were derived using the mean plus 3 standard deviations above a

negative control panel [35]. For the remaining antigens we used finite mixture models to fit Gaussian distributions for the $\log_{10}$ transformed MFI-bg values [13,41] and determined the seropositivity cutoffs using the mean plus three standard deviations of the first component. When estimating seropositivity cutoffs using mixture models, we restricted the population to children age 0 to 2 years at the exam date to ensure a sufficient number of unexposed children (S1 Fig) [13].

For the descriptive seroprevalence analyses we included all sampled children (aged 0 to 9 years old). In the analysis of antibody response and distance to water source we restricted the age range to 0 to 3 years for most enteropathogens because there was almost no outcome heterogeneity above age 3, consistent with other enteropathogen serology in cohorts from low-resource settings [13]. For pathogens with presumed lower transmission based on more slowly rising age-dependent seroprevalence (*C. trachomatis* and *S. enterica*) we used the full age range (0 to 9 years) [12]. All age ranges were pre-specified.

The relationship between age and seroprevalence is usually non-linear and varies by infection dynamics. Therefore, we sought a flexible modeling approach that does not impose assumptions on the functional form of the relationship between age and seropositivity. First, we used a stacked ensemble machine learning algorithm called "super learner" that combines predictions from multiple algorithms to ensure the best estimate of the age-dependent seroprevalence [42]. We included the following algorithms in the library: the simple mean, generalized linear models (GLMs), locally weighted regression (loess), generalized additive models with natural splines, and random forest. The super learner algorithm weights each member of the library so that the combined prediction from the ensemble minimizes the cross-validated mean squared error. Ensemble fits of age-antibody curves do not converge at the standard $n^{1/2}$ rate so pointwise confidence intervals are difficult to estimate [43]. We therefore also estimated the age-dependent antibody curves using a cubic spline for age within a generalized additive model (GAM) [44]. We estimated approximate simultaneous confidence intervals around the curves using a parametric bootstrap of the variance-covariance matrix of the fitted model parameters [45,46].

To estimate differences in seroprevalence according to distance to the nearest water source, we used doubly robust targeted maximum likelihood estimation (TMLE) with influence-curve based standard errors that treated clusters as the independent unit of analysis. We calculated prevalence differences comparing the prevalence in the furthest (fourth) quartile of distance to the nearest water source to the prevalence in the nearest (first) quartile. This comparison was prespecified. When comparing children living in the two quartiles we were restricted to roughly half of the sample size. We included the same algorithms as above in the TMLE super learner library to adjust for age and other potential confounders. For the random subset with a household survey, we adjusted for socio-economic status (SES) using quintiles of an asset index score calculated using a principal component analysis[47] of the following variables: if the household had electricity, the animals owned and species, education of the head of household and if someone in the household owned a radio. We also compared differences in quantitative antibody response according to distance quartile using the same approach.

The analysis plan was pre-specified and is available through the open science framework (osf.io/2r7tj). All analyses were done in *R* (version 3.4.2).

## Results

We collected dried blood spots from 2267 children residing in 40 communities between January and March of 2016. The median age was 5 years (IQR 3–7); 51.6% (1169/2267) of children were female. The median distance to the nearest water source was 448 meters (IQR 268–719).

The majority of children, 56.9% (1291/2267), lived in households whose nearest water source was unprotected. Household demographic information was available for 755 children. In this subset, 8.7% (66/755) of children lived in households with electricity, 10.1% (76/755) lived in households with a radio, 0% (0/761) lived in households with a mobile phone, 84.4% (637/755) lived in households that owned animals. For the majority of households (85.2% (643/755)), the primary occupation was agricultural work. (Table 1).

The seroprevalence among 0–9 year-olds was 43.1% (95% CI: 38, 48.4) for *C. trachomatis*, 27.5% (95% CI: 23.6, 31.6) for *S. enterica*, 70.3% (95% CI:67.7, 72.8) for *E. histolytica*, 53.9% (95% CI: 51.8, 56.0) for *G. intestinalis*, 95.6% (95% CI: 94.4, 96.5) for *C. jejuni*, 76.3% (95% CI: 74.1, 78.4) for ETEC and 94% (95% CI: 92.8, 94.9) for *C. parvum*. Seroprevalence increased with age with marked differences across pathogens. The age-dependent seroprevalence of *G. intestinalis declined after age 2*. (Fig 1). For ETEC, *E. histolytica*, *C. parvum*, *C. jejuni* and *G. intestinalis*, over 70% of children were positive at age 2 years. The age-dependent seroprevalence slopes were less steep for both *C. trachomatis* and *S. enterica*; by age 9 over 60% of children were seropositive for *C. trachomatis* and over 40% of children were seropositive for *S. enterica*. Seropositivity for more than 1 pathogen was common (Fig 2). At age 2 years, the median number of pathogens to which a child was seropositive was 4 (IQR 3–5), increasing to 5 (IQR 4–6) by age 4 years.

There was no indication for trend in community-level seroprevalence by community-level median distance to the nearest water source; however, there was considerable variability on community-level seroprevalence for some pathogens (*C. trachomatis*, *G. intestinalis*, *E. histolytica* and *S. enterica* (Fig 3)). The between-community variance in seroprevalence was highest for *C. trachomatis* (SD .20) and *S. enterica* (SD 0.13). More community-level heterogeneity was apparent among young children (under 3) compared with older children, the exceptions being *C. parvum* and *C. jejuni* which both had very high seroprevalence even among young children. Correlation between community-level seroprevalence illustrated variation in co-occurrence (S2 Fig). There was indication for pair-wise correlation in community level seroprevalence between *C. trachomatis* and *E. histolytica*, ETEC and *S. enterica*, *C. jejuni* and C. parvum, and C. parvum and E. histolytica (Pearson correlation > 0.3) (S2 Fig).

**Table 1. Population characteristics in overall population and subset with household survey.**

|  | Overall sample | Subset with household survey |
|---|---|---|
| n() children | 2267 | 755 |
| n() communities | 40 | 40 |
| Median age (IQR) | 5 (3–7) | 5 (3–7) |
| Female | 1169 (51.6%) | 377 (49.9%) |
| Median distance (meters) to nearest water source (IQR) | 473 (268–719) | 482 (268–737) |
| **Nearest water source** | | |
| Surface water | 1195 (52.7%) | 426 (56.4%) |
| Unprotected dug well | 76 (3.4%) | 26 (3.4%) |
| Protected spring | 548 (24.2%) | 173 (22.9%) |
| Protected dug well | 448 (19.8%) | 130 (17.2%) |
| **Household Characteristics** | | |
| Primary occupation of HH: agricultural work | • | 643 (85.2%) |
| Household has electricity | • | 66 (8.7%) |
| Household has radio | • | 76 (10.1%) |
| Household owns animals | • | 637 (84.4%) |
| Household has mobile phone | • | 0 (0%) |

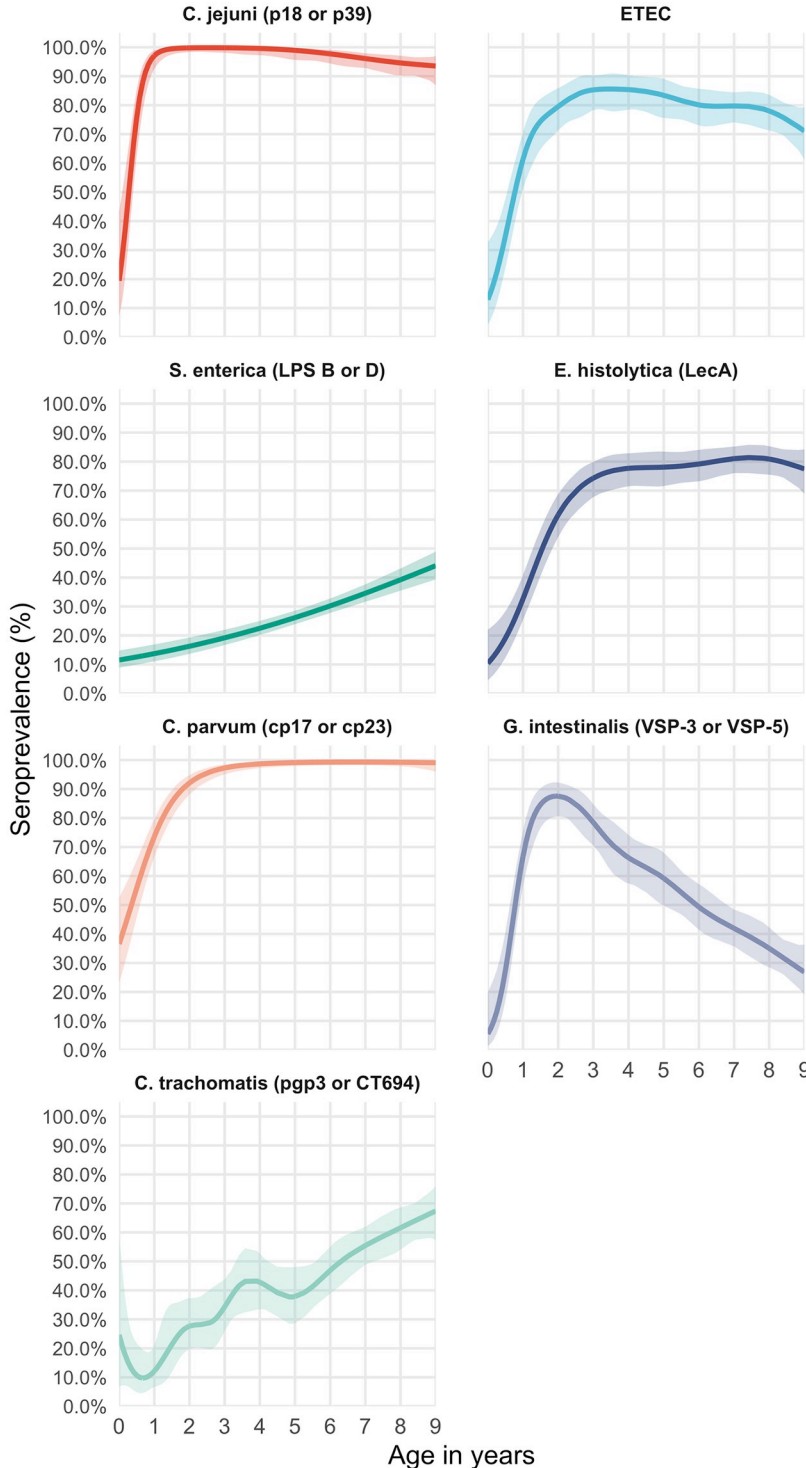

**Fig 1. Age-dependent seroprevalence of trachoma and enteropathogens in the Amhara region of Ethiopia.** Age-dependent seroprevalence curves were fitted using generalized additive models (GAM) with a cubic spline for age. Seropositivity cutoffs were derived using ROC curves, if available, or by fitting finite mixture models (S1 Fig). Seropositivity cutoffs could not be estimated for V. cholerae in this study, so seroprevalence curves are not shown. For pathogens with more than one antigen, positivity to either antigen was considered positive. IgG response measured in multiplex using median fluorescence units minus background (MFI-bg) on the Luminex platform on 2267 blood samples from 2267 children.

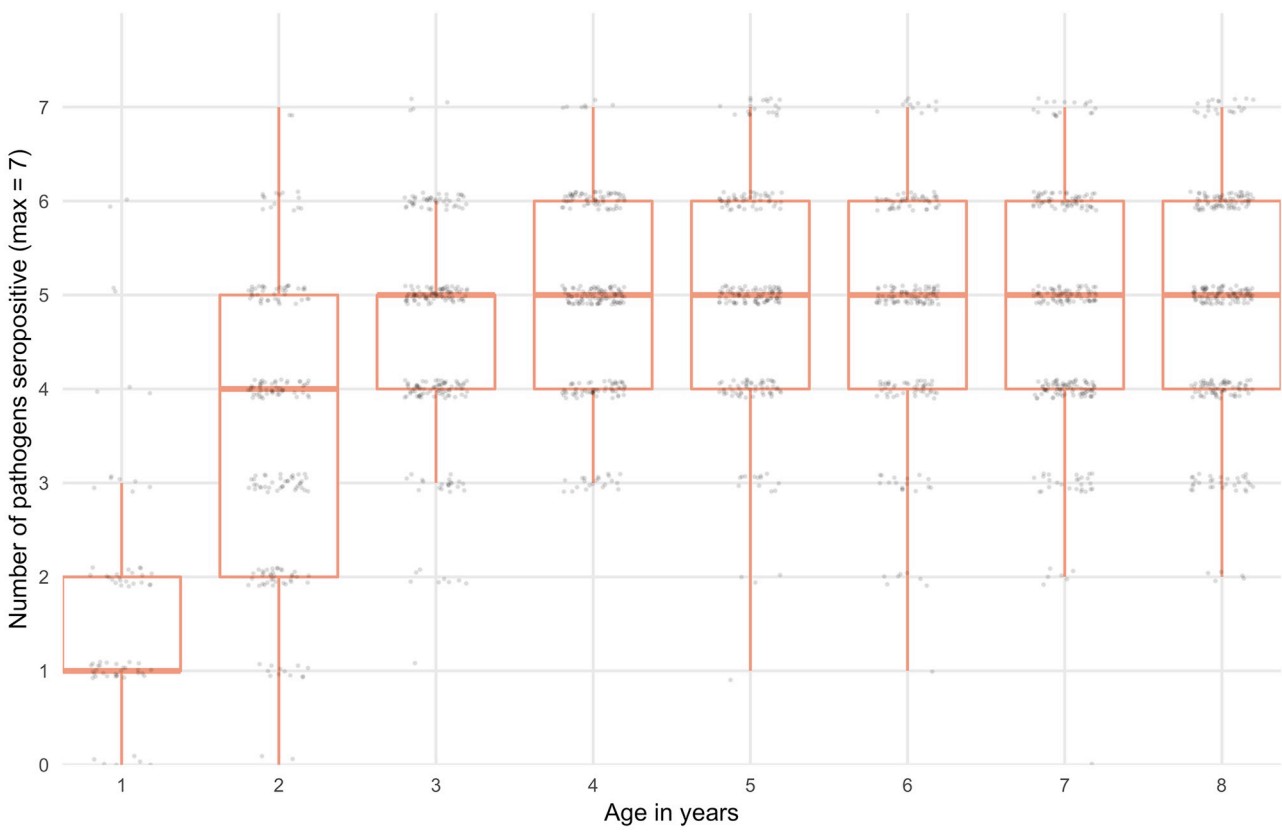

**Fig 2. Seropositivity for more than 1 pathogen by age.** Boxplot depicts median, upper and lower quartiles. Seropositivity cutoffs were derived using ROC curves, if available, or by fitting finite mixture models (S1 Fig). IgG response measured in multiplex using median fluorescence intensity minus background (MFI-bg) on the Luminex platform on 2267 blood samples from 2267 children.

Children in the quartile living farthest from any water source had a 12% (95% CI: 2.6, 21.4) higher seroprevalence of *S. enterica* and a 12.7% (95% CI: 2.9, 22.6) higher seroprevalence of *G. intestinalis* compared to children living in the nearest quartile (Table 2). Quantitative antibody levels demonstrated the same pattern for *S. enterica*, with antibody levels for *S. enterica* LPS group D 0.32 (95% CI: 0.13, 0.52) $\log_{10}$ MFI-bg units higher among children living in the furthest quartile from water compared to children living in the nearest quartile (p = 0.001) (S1 Table). Quantitative antibody levels for ETEC and *G. intestinalis* were slightly higher among children living in the furthest quartile, but the differences were not statistically significant.

In the subset of children with household-level data, point estimates were similar but there was no longer a statistically-significant association between distance to the nearest water source and seroprevalence for *S. enterica*, ETEC or *G. intestinalis* in the unadjusted or SES-adjusted analysis largely due to the smaller sample size and wider confidence intervals (Table 2).

## Discussion

This study found high exposure to *C. trachomatis* and enteric pathogens among children residing in rural areas of the Amhara region of Ethiopia. Seroprevalence was age-dependent, with over 70% of children seropositive for ETEC, *E. histolytica*, *C. parvum*, *C. jejuni* and *G. intestinalis* at age two years. Age-dependent seroprevalence rose more slowly for *S. enterica* and *C. trachomatis*, suggesting lower transmission compared with the other enteropathogens. Still, at

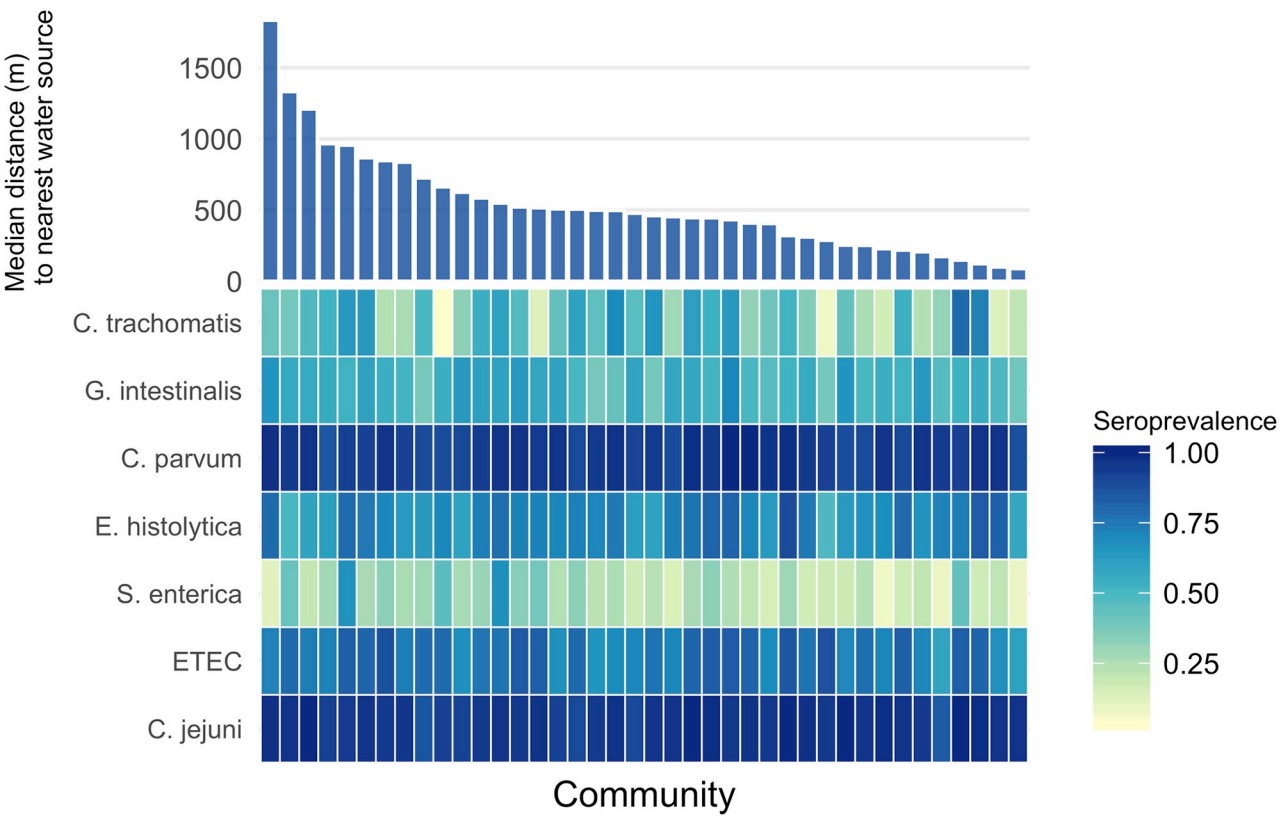

**Fig 3. Variation in seroprevalence by community and distance to the nearest water source.** Heatmap of community-level seroprevalence, darker colors indicate higher seroprevalence. Communities are sorted by median distance to the nearest water source, from furthest to nearest. Seropositivity cutoffs were derived using receiver operating characteristic (ROC) curves, if available, or by fitting finite mixture models (S1 Fig). For pathogens with more than one antigen, positivity to either antigen was considered positive. IgG response measured in multiplex using median fluorescence intensity minus background (MFI-bg) on the Luminex platform on 2267 blood samples from 2267 children aged 0 to 9 years.

age 9 years, over 60% of children were seropositive for *C. trachomatis* and over 40% of children were seropositive for *S. enterica*.

Unlike for other pathogens in the study, G. intestinalis seroprevalence declined after age two years. Giardia has been shown to exhibit increasing infection prevalence with age in other cohorts in low-resource settings with a high proportion of asymptomatic infections [48], suggesting that the IgG response is weaker at older ages despite infection. The precise immunological mechanism for lower mean IgG levels among older ages is not currently known, but the phenomena has been observed in multiple other cohorts. For example, Arnold et al. demonstrated declining mean IgG with age for Giardia (VSP-3, VSP-5), ETEC (LTB) and Campylobacter (p18, p39) in cohorts from Haiti and Kenya [13]. Age-dependent antibody kinetics in that study suggest that much of the decline of mean IgG with age for these pathogens is likely due to acquired immunity, which results in either lower rates of infection, or more likely, if children are infected they experience less severe disease and potentially a less robust IgG boost.

Use of a multiplexed immunoassay allowed us to expediently identify that seropositivity to more than one pathogen was common in the Amhara region of Ethiopia and that, by age three, most children were seropositive for five of the seven pathogens under investigation. Similarly, we were able to identify notable correlation in seroprevalence between some pathogens (for example, *C. parvum* and *E. histolytica)* at the community level. The seroprevalence of *G. intestinalis* and *E. histolytica* in this study was substantially higher than the prevalence reported

**Table 2. Seroprevalence according to distance to the nearest water source.**

| Pathogen | n() sero-positive and seroprevalence (%) according to distance to the nearest water source (quartiles) | | | | Full Dataset | | Subset* | | Subset* adjusted for SES | |
| | Q1 | Q2 | Q3 | Q4 | | | | | | |
| | n = 566 | n = 568 | n = 564 | n = 565 | | | | | | |
|  | (Age <3, n = 134) | (Age <3, n = 138) | (Age <3, n = 123) | (Age <3, n = 125) | Prevalence difference Q4 to Q1 | P-value | Prevalence difference Q4 to Q1 | P-value | Prevalence difference Q4 to Q1 | P-value |
|---|---|---|---|---|---|---|---|---|---|---|
| S. enterica (LPS B or D) | 112 (20.1%) | 150 (26.5%) | 173 (30.9%) | 182 (32.3%) | 12% (2.6, 21.4) | 0.012 | 5.9% (-5, 16.7) | 0.289 | 5.7% (-4.8, 16.1) | 0.288 |
| C. trachomatis (pgp3 or CT694) | 235 (42.1%) | 235 (41.5%) | 242 (43.2%) | 258 (45.8%) | 3.5% (-6.9, 13.9) | 0.514 | 2.2% (-13, 17.4) | 0.776 | 3.9% (-11.6, 19.4) | 0.620 |
| C. jejuni (p18 or p39)** | 124 (93.2%) | 126 (91.3%) | 108 (87.8%) | 114 (91.2%) | -0.6% (-5.8, 4.5) | 0.806 | -5.1% (-10.3, 0.2) | 0.057 | -5.2% (-10.5, 0) | 0.052 |
| ETEC** | 80 (60.2%) | 93 (67.4%) | 75 (61%) | 81 (64.8%) | 6.4% (-7, 19.9) | 0.350 | 1.7% (-15.4, 18.9) | 0.843 | 2.3% (-18.3, 23) | 0.827 |
| E. histolytica (LecA)** | 66 (49.6%) | 61 (44.2%) | 42 (34.1%) | 54 (43.2%) | -5.7% (-20.4, 9.1) | 0.451 | -20% (-43.8, 3.8) | 0.100 | -20.3% (-46, 5.3) | 0.120 |
| C. parvum (cp17 or cp23)** | 104 (78.2%) | 114 (82.6%) | 92 (74.8%) | 91 (72.8%) | -3.9% (-16.5, 8.6) | 0.539 | -3.2% (-23.8, 17.3) | 0.757 | -10.5% (-38, 17.1) | 0.458 |
| G. intestinalis (VSP-3 or VSP-5)** | 86 (64.7%) | 90 (65.2%) | 78 (63.4%) | 94 (75.2%) | 12.7% (2.9, 22.6) | 0.011 | 8.3% (-8.6, 25.3) | 0.334 | 9.1% (-9.1, 27.2) | 0.328 |

All prevalence difference estimates are adjusted for age and account for variation in the standard error due to clustering by community.

*Subset = random 33% of households with socioeconomic status information

** Age restricted to 0–3 years

Quartile 1 (Q1): 11.4m–267m; Quartile 2 (Q2): 268m–472m; Quartile 3 (Q3): 473m–720m; Quartile 4 (Q4): 721–2906m

in studies using microscopy in the region. In one recent study of protozoan prevalence in the Amhara region, the single-stool prevalence of *Entamoeba spp. (histolytica and dispar)* by microscopy among three year old children was 7.1% [49]. However, differences between seroprevalence and prevalence by microscopy are expected given that IgG response integrates information over time and microscopy measures active presence and shedding. The seroprevalence of C. trachomatis identified in this study is consistent with the high burden of trachoma documented in the Amhara region [50].

Children living farther from a water source had higher seroprevalence of *S. enterica* and *G. intestinalis*. The absence of heterogeneity in seroprevalence in this high transmission setting may have masked other potential relationships between exposure to enteric pathogens and distance to water. For example, among children 0 to 3 years old, the seroprevalence of *C. parvum* and *C. jejuni* were both very high (77% and 91% respectively). In a sensitivity analysis restricted to children younger than 12 months, there was an indication that the quantitative antibody levels for children living in the farthest quartile of distance compared to the nearest quartile of distance were higher for *V. cholerae* toxin beta subunit, *C. parvum* cp17 and cp23. However, the differences among this age sub-group were not statistically significant; the statistical power was likely limited by the lower number of children in this subset.

We were likely underpowered to determine differences in seroprevalence adjusted for socio-economic status. In the random 33% subset of children with available household asset information, children living in the furthest quartile of distance still had a higher seroprevalence of *S. enterica* and *G. intestinalis*, however the differences were not statistically significant.

There were several limitations of this study with respect to how the nearest water source was measured. First, we measured absolute Euclidean distance rather than walking distance or time it takes to collect water. The study site region has tremendous gradation in altitude, with

many high plateaus and steep valleys. In some cases, the distance to the nearest water source may not reflect the time it would take to ascend, descend or otherwise traverse the terrain. Future studies may consider alternative methods for calculating distance that accommodate land type and elevation changes. Second, we did not ask household which water source they were using. Households may use water sources that are further away via linear distance because of taste preference, ease of access, water source type or other reasons, namely terrain [51]. Third, the study site region is arid and there is variation in water availability by season. We simply measured the distance to the nearest water source at the time of the census and this may have not reflected a water source that was flowing and available at different times of the year. Third, we assumed that distance to the nearest water source was associated with the quantity of water used by the household. Future studies could use sensors or measure the reported number of jerrycans used over time to more precisely measure water quantity. All of the above scenarios may have introduced non-differential misclassification of the exposure, which could bias associations towards the null. Finally, we opted to measure distance to the nearest protected or unprotected water source to evaluate the effect of water quantity on enteropathogen and *C. trachomatis* transmission rather than water quality. An alternative approach would be to evaluate the effect of water quality on enteropathogen transmission would be to assess the type of water source that was used by each household, measure the distance to that source and then evaluate associations between distance, water source type and seroprevalence, ideally tracking microbiological water quality \ longitudinally.

The association between water quality and risk of exposure and susceptibility to infections is subject to many potential confounding variables that we were unable to measure such as household and community level hygiene and sanitation practices and water treatment and storage practices. Future studies should consider measuring and evaluating these variables.

Another limitation of this study was the difficulty in determining seropositivity cut-offs for several of the antigens. The enteropathogens in particular pose a challenge. We were unable to determine reasonable cutoffs for *C. parvum* and *V. cholerae* using mixture models and had to discard *V. cholerae* from the seroprevalence analysis without a corresponding external negative control cutoff. Analyzing quantitative antibody levels is an alternative to seroprevalence that may retain the higher resolution needed in high-transmission settings [12]. When we evaluated differences in quantitative antibody levels according to distance to the nearest water source, the results were consistent with the seroprevalence findings for *S. enterica* LPS group B; quantitative antibody levels were also higher for ETEC and *G. intestinalis* but the differences were not statistically significant.

In conclusion, in this large population-based study of young children in the Amhara region of Ethiopia we document high transmission of *C. trachomatis*, *G. intestinalis*, *C. parvum*, *E. histolytica*, *S. enterica*, *C. jejuni* and ETEC. Children living furthest from a water source had higher seroprevalence of *S. enterica* and *G. intestinalis* compared to children living closest to a water source. Serology was a useful approach to measure exposure to *C. trachomatis* and multiple enteropathogens. Our findings indicate the improving water quantity, through minimizing the distance to water collection, may reduce enteric pathogen transmission in settings such as Amhara with extreme water scarcity.

## Supporting information

**S1 Fig. Distribution of IgG antibody response among children <24 months with ROC and mixture model cutoffs.** IgG antibody response measured in multiplex using median fluorescence units minus background (MFI-bg) on the Luminex platform. Population restricted to children <24 months to derive cutoffs (n = 317). Vertical lines mark seropositivity cutoffs

based on external negative controls (solid) and finite Gaussian mixture models (dash). For Chlamydia trachomatis pgp3 & CT694 cutoffs were derived using receiver operating characteristic (ROC) curves, for Cryptosporidium parvum Cp17 & Cp23 cutoffs were derived using a standard curve and for Giardia intestinalis VSP-3 & VSP-5 and Entamoeba histolytica LecA cutoffs were derived using the mean plus 3 standard deviations above a negative control panel. (TIFF)

**S2 Fig. Community-level correlation in seroprevalence.** Correlation between the mean community seroprevalence depicted with circles, greater circle area represents higher correlation. For pathogens with more than one antigen, positivity to either antigen was considered positive. IgG response measured in multiplex using median fluorescence units minus background (MFI-bg) on the Luminex platform on 2267 blood samples from 2267 children aged 0 to 9 years. (TIFF)

**S1 Table. Quantitative antibody levels by distance quartile and differences comparing Quartile 4 to Quartile 1.**
(DOCX)

## Acknowledgments

We gratefully acknowledge the study participants for their valuable time. Purified Cp17, Cp23, VSP-3, VSP-5, p18, and p39 antigens were kindly provided by Jeffrey Priest (US CDC), and LecA antigen was kindly provided by William Petri (University of Virginia) and Joel Herbein (TechLab).

## Disclaimer

The findings and conclusions in this report are those of the authors and do not necessarily represent the official position of the Centers for Disease Control and Prevention. Use of trade names is for identification only and does not imply endorsement by the Public Health Service or by the U.S. Department of Health and Human Services.

## Author Contributions

**Conceptualization:** Kristen Aiemjoy, Diana Martin, Jeremy D. Keenan, Benjamin F. Arnold.

**Data curation:** Kristen Aiemjoy, Benjamin F. Arnold.

**Formal analysis:** Kristen Aiemjoy, Sarah Gwyn, Diana Martin, Benjamin F. Arnold.

**Funding acquisition:** Jeremy D. Keenan.

**Investigation:** Kristen Aiemjoy, Jeremy D. Keenan, Benjamin F. Arnold.

**Methodology:** Kristen Aiemjoy, Solomon Aragie, Dionna M. Wittberg, Sarah Gwyn, Diana Martin, Jeremy D. Keenan, Benjamin F. Arnold.

**Project administration:** Solomon Aragie, Dionna M. Wittberg, Zerihun Tadesse, E. Kelly Callahan.

**Resources:** Diana Martin.

**Software:** Benjamin F. Arnold.

**Supervision:** Solomon Aragie, Dionna M. Wittberg, Diana Martin, Jeremy D. Keenan, Benjamin F. Arnold.

**Validation:** Sarah Gwyn.

**Visualization:** Kristen Aiemjoy, Benjamin F. Arnold.

**Writing – original draft:** Kristen Aiemjoy, Benjamin F. Arnold.

**Writing – review & editing:** Solomon Aragie, Dionna M. Wittberg, Zerihun Tadesse, E. Kelly Callahan, Sarah Gwyn, Diana Martin, Jeremy D. Keenan.

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
