## [Decision Letter · Decision Letter 0]

2 Jun 2020

Dear Dr. Aiemjoy,

Thank you very much for submitting your manuscript "Seroprevalence of antibodies against Chlamydia trachomatis and enteropathogens and distance to the nearest water source among young children in the Amhara Region of Ethiopia" for consideration at PLOS Neglected Tropical Diseases. As with all papers reviewed by the journal, your manuscript was reviewed by members of the editorial board and by several independent reviewers. The reviewers appreciated the attention to an important topic. Based on the reviews, we are likely to accept this manuscript for publication, providing that you modify the manuscript according to the review recommendations. 

Sincerely,

Jeremiah M. Ngondi, MB.ChB, MPhil, MFPH, Ph.D

Associate Editor

Elsio Wunder Jr

Deputy Editor

Reviewer's Responses to Questions

**Key Review Criteria Required for Acceptance?**

**Methods**

-Are the objectives of the study clearly articulated with a clear testable hypothesis stated?

-Is the study design appropriate to address the stated objectives?

-Is the population clearly described and appropriate for the hypothesis being tested?

-Is the sample size sufficient to ensure adequate power to address the hypothesis being tested?

-Were correct statistical analysis used to support conclusions?

-Are there concerns about ethical or regulatory requirements being met?

Reviewer #1: Methods: It would be helpful to know a bit more about the study location, not just that it’s rural. Are pastoral activities common? Climate? Extreme water scarcity is mentioned in the last sentence of the discussion, but more could be said in the methods about the context. 

Is “community” the cluster unit? I assume so, but please clarify. 

Study population, second paragraph, first sentence:

“in December 2015, approximately one month before the baseline examination visit” please clarify, one month before the baseline examination visits began, as it appears the baseline took 4 months to conduct. This time gap is important to note, as the authors later highlight that the list of sources according to may not reflect sources actually used in the month(s) prior to the dried blood spot collection. 

Study population, second paragraph, last sentence: Though not the focus of this study, please clarify how the primary outcome for the overall trial was determined? And please make more clear in this sentence that the sample size was therefore not calculated for the intention of performing the analyses presented in this manuscript (correct?). 

Measurements, dried blood spots paragraph – please clarify length of time samples stored at -20C in total before sample analysis. Any concern of sample degradation? 

Distance to water. When community leaders listed all sources of water, did they also describe type of source, name? Was there much heterogeneity in the number of sources per village, any effort made to validate community leader report to actual on-the-ground use of water sources by households? Please clarify, given potential issues of recall reliability and thoroughness. How did the census workers know they had reached the water source described by the community leaders? Were they accompanied to the source? 

GPS point -- was a minimum accuracy achieved before recording the waypoint? (e.g. <10m?)

“and thus used distance to the nearest water source (improved or unimproved) for the analysis.” So basically making the assumption that this is a proxy for water quantity? This should be included as a limitation. 

Statistical analysis - 

“For the remaining antigens we used finite mixture models to fit Gaussian distributions for the log10 transformed MFI-bg values [10,22] and determined the seropositivity cutoffs using the mean plus three standard deviations of the first component.” Does the Priest et al. paper use finite mixture models, ? Please clarify. If no, is there any paper you can cite to justify this approach.

“consistent with other enteropathogen serology in cohorts from low-resource settings” The authors cite the Priest et al. 2006 paper, but imply this finding is found in multiple cohorts. Please cite at least an additional paper demonstrating this consistency, particularly one that covers additional pathogens besides the Cryptosporidium species reported in the Priest et al. 2006 paper. 

“For pathogens with presumed lower transmission based on more slowly rising age-dependent seroprevalence (C. trachomatis and S. enterica)” 

 Is there a paper that can be cited to justify this presumption? 

“Among the 33% of children whose household was randomly selected for inclusion in the household survey, we adjusted for socio-economic status (SES) using an indicator variable calculated using a principal component analysis.” Meaning this is a sensitivity test? Please provide more details on the variables used to construct the PCA, perhaps as supplemental material. Did you calculate quintiles?

Reviewer #2: The objectives and methods of this study were very well thought out and clearly articulated. The methods were appropriate to fulfill the two primary objectives. Because this is a secondary analysis of data from another study, there was not sufficient statistical power to control for socio-economic status. However, this limitation was clearly outlined. 

The predictor of interest, distance from water, was calculated simply as the median linear distance from households in a community to the nearest water source. There are many limitations to this definition, and they are adequately explained in text. However, although the authors clearly stated their rationale for ignoring water quality in this analysis, I think that it would improve the paper to take water quality into account.

The analysis does not take into account whether water sources are improved or unimproved. Given that information on whether water sources are improved or unimproved is available, the authors should test whether access to improved water is independently associated with seroprevalence, and whether it is a possible confounder in the relationship between distance to water and seroprevalence (is access to improved water associated with both distance from water and seroprevalence of C trachomatis and/or GI pathogens of interest?). It is plausible that transmission of C trachomatis is associated primarily with quantity of water, but transmission of GI pathogens can be linked to poor quality water.

Reviewer #3: Methods / Study population: It would be helpful to clarify how childrens ages were recorded (years, months, days?).

Methods / Covariates: Please provide the rationale behind only assessing socioeconomic status in one-third of households. Also, please provide detail or references to how the household status questionnaires were designed and how the factors presented in the results (such as having a mobile phone or a radio) compare to socioeconomic status and human behaviour. 

Methods / Laboratory methods: Please add a sentence describing why these particular enteropathogens were selected for study. 

Methods / Laboratory methods: Please provide more information about the “two positive controls”. Are these serum samples with known exposure to all pathogens in the panel, or a composite reference of multiple sera? How were they defined as positive? A citation would suffice if this has been published elsewhere. 

Methods / Statistical analysis / Calculation of age-dependent seroprevalence section: For the benefit of the non-statistician audience of this paper, suggest adding a few sentences describing the analysis approach in lay terms and why it is beneficial over more simplistic approaches.

**Results**

-Does the analysis presented match the analysis plan?

-Are the results clearly and completely presented?

-Are the figures (Tables, Images) of sufficient quality for clarity?

Reviewer #1: “51.3% (1169/2267) of children were female” 

Please double-check this – I get 51.6% when using 1169/2267 (as reported in Table 1)

Table 1 – I suggest reporting meters to nearest whole meter given GPS accuracy issues. 

Surface water types: preferable to report according to JMP classifications if possible, e.g. surface water instead of open water, unprotected dug well instead of unprotected well. 

Table 1 – animal ownership; possible to separate out by cows, chickens, etc? Or those kept in compound? 

Possible to include anything related to handwashing or sanitation facilities? Occupation? Education level? 

Table 1 – for subset with household survey, are these nearest water sources still based on the list generated by the community leaders? Or are these water sources according to household survey answers? If the latter, did these source types match up with the source type indicated according to the method you used for determining nearest water source? 

In the household survey, did you collect round-trip travel time to go to source, collect water, and return? 

Figure 1 - I might have missed it, but what might explain the waning seroprevalence of G. intestinalis as age increases after 2 years? Does it become a long-term sub-clinical infection with a corresponding reduction in antibody response? It appears much different than the other pathogens, and I think it deserves some more discussion. Also, regarding age-dependent seroprevalence patterns for S. enterica and C. trachomatis in contrast to C. jejuni, ETEC, E. histolytica, and C. parvum -- could this variation be somehow linked to child and/or caretaker behavior, or household-specific factors, in addition to low vs. high transmission more generally? Evidence of this phenomenon in other studies? Also, the benefits of delayed seroconversion for an individual (e.g. growth, stronger immune system once pathogen is eventually encountered, improved cognitive development?) might be worth highlighting, as any associations become a little less noteworthy to policymakers if everyone is going to seroconvert eventually. However, is there any evidence that delayed seroconversion could actually result in a stronger immune response at an older age resulting in more severe illness? 

Figure 1 caption: “For pathogens with more than one antigen, positivity to antigen was considered positive.” 

Revise to state “…positivity to EITHER antigen was considered positive”, correct? Same for Figure 3 caption, and Supplemental Figure 2 caption. 

“There was no indication for trend in community-level seroprevalence by community-level median distance to the nearest water source” 

How much variability was there within a community to nearest water source? You’ve presented the medians, but is there anything else you could show, and might that be predictive of village seroprevalence heterogeneity among children <3? 

Was there any association between water source type and distance to water source? SES and distance? Was there any association between animal ownership and distance? Would that be something to adjust for? Was animal ownership part of the SES index? Could those who live futher from water sources have more animal exposures?

Why not adjust for crowding, sanitation and hygiene indicators if testing hypothesis of water source distance resulting in higher exposure to C. trachomatis and enteric pathogens?

Reviewer #2: The results are very clearly presented and match the described methods. The figures were exceptionally well done and clear.

Reviewer #3: (No Response)

**Conclusions**

-Are the conclusions supported by the data presented?

-Are the limitations of analysis clearly described?

-Do the authors discuss how these data can be helpful to advance our understanding of the topic under study?

-Is public health relevance addressed?

Reviewer #1: Discussion, second paragraph: can you infer active infection from higher antibody count? 

“To evaluate the effect of water quality on enteropathogen transmission, distance to the nearest protected water source may have been a more appropriate exposure.” I fail to see how calculating distance to the nearest protected water source vs all water sources as was done would help in evaluate effect of water quality on enteropathogen transmission, since it would be hard to make the case that households were overlooking nearer water sources of worse quality for non-drinking purposes. Wouldn’t this just increase the distance to water source only for those households that happened to have an unimproved as their nearest source? Also, improved water sources do not necessarily imply safe water provision, and at least in one recent paper, handwashing with poor quality water is still effective. Perhaps I’m missing something, but this seems like a stretch. I would say, it would be better to actually measure the water quality (ideally longitudinally). 

“Analyzing qualitative antibody levels is an alternative to seroprevalence that may retain the higher resolution needed in high-transmission settings [9].” 

You mean quantitative, not qualitative, correct? Please revise. 

“The study site region has tremendous gradation in altitude, with many high plateaus and steep valleys.” There are methods for doing ansiotropic models for travel time that utilize land types and elevation changes, usually in context of health facility accessibility, but could be applied to water source accessibility, e.g. Access MOD https://www.accessmod.org/ It would be worth mentioning future studies could use more sophisticated approaches to overcome the limitations of using Euclidean distance as a proxy for water source access. On the other hand, it might be better than reported travel time, and can be a decent proxy for route distance (Ho et al. 2014 https://doi.org/10.2166/wh.2013.042)

Additionally, it might be worth mentioning alternative approaches to measuring water quantity, since that seems central to your hypothesis; e.g. could use sensors or measure reported number of jerricans fetched by size, per day or per week. Also worth noting that water quantity and usage for hygiene, along with water source availability as you’ve already pointed out, likely varies by season. Others have made the argument that focusing on household water quantity and amount used for hygiene purposes may be more predictive of trachoma and other disease risk. (Stelmach and Clasen 2015, Altherr 2019)

Supplemental Figure 1 caption – correct typos (recever and Entomoeba)

Supplemental Figure 2 caption – perhaps I missed it, but I was a little surprised to see “2328 blood samples from 2328 children aged 0 to 9 years” when everywhere else it was “2267 blood samples from 2267 children” Please clarify.

Reviewer #2: The conclusion section is well-written and includes a clear description of limitations.

Reviewer #3: Discussion: “The seroprevalence of trachoma” typo, please change to sero prevalence of C. trachomatis.

Discussion: Can authors comment on the decrease in age specific seroprevalence on Giardia between the ages of 2 and 9 years? And the decrease in ETEC seroprevalence between 5 and 9 years? Is this seroreversion?

**Editorial and Data Presentation Modifications?**

Reviewer #1: Minor comments are as follows, perhaps a little challenging to follow without line numbering, but hopefully doable: 

Last sentence of author summary: something missing here. “the relationship [OF?] water, sanitation and hygiene related exposures [TO?] disease transmission”

Background, paragraph 1, first sentence: do either reference 1 or 2 describe burden of diarrhea? Consider citing GBD or other review as background for diarrhea. Also, are diarrhea and trachoma causes of morbidity, or the actual morbidity? Wouldn’t causes be the upstream determinants resulting in these two conditions? Consider rephrasing. 

Background, paragraph 1, last sentence: Is this the correct reference for the statement? Please consider citing additional papers that address water quantity, use for hygiene, and distance from water source. 

Eg. https://doi.org/10.1016/j.wrr.2014.04.001

Also specific to Ethiopia: Gibson and Mace 2006https://journals.plos.org/plosmedicine/article?id=10.1371/journal.pmed.0030087

Also, water fetching time and trachoma risk: Altherr 2019 https://link.springer.com/article/10.1186/s13071-019-3790-3

Consider also this review by Cassivi 2019 https://doi.org/10.1016/j.ijheh.2019.06.011 and/or Overbo 2016: https://doi.org/10.1016/j.ijheh.2016.04.008 … note distinction between water-washed and waterborne infections, which you may want to mention with respect to the panel you’ve examined. 

“a longer window to identify exposed individuals” Just curious -- does exposure to the pathogens under study necessarily result in seroconversion, development of antibodies, do they wave over time? Or might the dose influence likelihood of infection and seroconversion? Is there any literature that you can cite on this, perhaps in the discussion, could this be a limitation? Or maybe “infected individuals” could be more accurate than “exposed individuals”? 

Background , second paragraph, 4th sentence. Antibody response generally? Or quantitative antibody response more specifically, i.e. has the potential to differentiate current (regardless of symptomology?) vs past infection? A little confusing what is meant by “enumerates”… I’m not sure if you’ve provided sufficient evidence to back up this claim, and I don’t believe the Lammie 2012 paper discuses symptomatic vs asymptomatic infections and corresponding antibody response. Please clarify and/or rephrase.

Reviewer #2: 1. It was not mentioned in the methods section that the community-level distance to water was defined as median distance to water of surveyed households in the community. This should be added. 

2. The sentence "There was indication for correlation between C. trachomatis and E. histolytica, ETEC, C. jejuni and S. enterica (Pearson correlation > 0.3)" (first full sentence p 12) is difficult to understand. The authors should clarify whether this sentence is referring to the pairs of pathogens listed as correlated in Supplemental Figure 2 and, if so, to which pairs of pathogens it is referring.

3. It is mentioned in text that seroprevalences of ETEC and C trachomatis were higher in the fourth quartile compared to the first quartile but the results are not statistically significant (second sentence, last paragraph, p 12). Although this is a true assertion, it is also true that the seroprevalences of E histolytica and C parvum were higher in the first quartile compared to the fourth quartile, but this was also not statistically significant. The authors should either remove mention of results that were not statistically significant or make equal mention of results that do and do not support their hypothesis. Alternatively, they could provide an explanation for why the ETEC and C trachomatis results were more noteworthy than the E histolitica and C parvum results.

4. In the third sentence in the last paragraph on page 16, "qualitative" was used when I believe that they authors meant to say "quantitative."

5. In the second sentence of the first paragraph on page 17, ETEC was mentioned as a pathogen for which seroprevalence was higher in the 4th quartile versus the 1st. Given my previous comments, I think this mention should be removed.

Reviewer #3: Abstract / Methods: “1-9 years” typo? Methods section says 0-9.

Author summary: “Trachoma, and infection of the eye” typo

Background: “Increased access to water for food preparation and washing of hands, faces, and clothing is hypothesized to reduce transmission of both infectious diarrhea and C. trachomatis [3–6].” There might be a better reference than the pinsent one for influence of water access on transmission of C. trachomatis.

**Summary and General Comments**

Reviewer #1: Thank for the opportunity to review this paper. It was very well written and I found it an interesting and enjoyable read. 

My comments are fairly minor, although in general, I think the authors need to provide more justification for the hypothesis and the biologic plausibility, paying particular regard to each of the pathogens under study and how they may differ according to transmission patterns, risk factors, and relation to water quantity. With a relatively crude indicator of water quantity (nearest source) assumed to correspond to water source distance, further assumed to be the nearest source used by that particular house according to a community leader’s tally, the potential role of unmeasured confounders should at least be noted. Potentially relevant determinants of both risk of exposure and susceptibility to infections could include water source reliability, water fetching and animal tending responsibilities, soil contact/ingestion, child feces management, household and community-level hygiene and sanitation practices, water treatment and storage practices, nutrition status, and other comorbidities. In future work, it could also be helpful to look for spatial clustering of the diseases of interest, particularly among children less than 3 years of age before seroprevalence (typically) plateaus, though I believe that is likely beyond the scope of this paper. I also think that the multiplicity of benefits (time-savings, improved safety, reduced musculoskeletal injury, stress, etc.) that comes with safe and more consistent and proximal water sources should be highlighted in the closing remarks of the discussion, as potential reductions of pathogen risk shouldn’t be considered in isolation. In short, this is an important piece of work and I applaud the authors’ efforts.

Reviewer #2: I appreciate the opportunity to review this manuscript. It describes an important study with results that have the potential to be used for advocacy for better water access in Ethiopia and elsewhere. The research was well-presented with clear and well-constructed tables and graphs. 

Other than the minor editorial concerns listed above, my only suggestion for the authors is to investigate whether water quality was a confounder to the relationship between distance to water and transmission of C trachomatis and/or GI pathogens. It appears that this analysis can be conducted with available data.

Reviewer #3: The authors present the data from serological analysis of ~2200 children in Ethiopia and assess the correlation between seropositivity and proximity to water source. In a smaller subset, they investigate more household variables related to socioeconomic status. The study is conducted in an area of high infectious disease burden, an appropriate context to utilise this type of tool, and the results are presented well. There are weaknesses to the manuscript, but these are for the most part acknowledged in the discussion. I believe the manuscript should be accepted for publication, pending the consideration of some minor points below.

PLOS authors have the option to publish the peer review history of their article (what does this mean?). If published, this will include your full peer review and any attached files.

Reviewer #1: No

Reviewer #2: No

Reviewer #3: No
---

## [Editor Report · Decision Letter 1]

27 Jul 2020

Dear Dr. Aiemjoy,

We are pleased to inform you that your manuscript 'Seroprevalence of antibodies against Chlamydia trachomatis and enteropathogens and distance to the nearest water source among young children in the Amhara Region of Ethiopia' has been provisionally accepted for publication in PLOS Neglected Tropical Diseases.

Best regards,

Jeremiah M. Ngondi, MB.ChB, MPhil, MFPH, Ph.D

Associate Editor

Elsio Wunder Jr

Deputy Editor

---

## [Editor Report · Acceptance letter]

26 Aug 2020

Dear Dr. Aiemjoy,

We are delighted to inform you that your manuscript, "Seroprevalence of antibodies against Chlamydia trachomatis and enteropathogens and distance to the nearest water source among young children in the Amhara Region of Ethiopia," has been formally accepted for publication in PLOS Neglected Tropical Diseases.

Best regards,

Shaden Kamhawi

co-Editor-in-Chief

Paul Brindley

co-Editor-in-Chief
